# Evidence on the formation of dimers of polycyclic aromatic hydrocarbons in a laminar diffusion flame

Alessandro Faccinetto [1✉], Cornelia Irimiea[2], Patrizia Minutolo [3], Mario Commodo [3], Andrea D'Anna [4], Nicolas Nuns[5], Yvain Carpentier[6], Claire Pirim [6], Pascale Desgroux[1], Cristian Focsa [6] & Xavier Mercier[1]

The role of polycyclic aromatic hydrocarbons (PAHs) in the formation of nascent soot particles in flames is well established and yet the detailed mechanisms are still not fully understood. Here we provide experimental evidence of the occurrence of dimerization of PAHs in the gas phase before soot formation in a laminar diffusion methane flame, supporting the hypothesis of stabilization of dimers through the formation of covalent bonds. The main findings of this work derive from the comparative chemical analysis of samples extracted from the gas to soot transition region of a laminar diffusion methane flame, and highlight two different groups of hydrocarbons that coexist in the same mass range, but show distinctly different behavior when processed with statistical analysis. In particular, the identified hydrocarbons are small-to-moderate size PAHs (first group) and their homo- and heterodimers stabilized by the formation of covalent bonds (second group).

[1] Univ. Lille, CNRS, UMR 8522, PC2A, F-59000 Lille, France. [2] DMPE, ONERA, Univ. Paris Saclay, F-91123 Palaiseau, France. [3] Istituto di Ricerche sulla Combustione, CNR, I-80125 Napoli, Italy. [4] Univ. Napoli Federico II, Dipartimento di Ingegneria Chimica, I-80125 Napoli, Italy. [5] Univ. Lille, CNRS, UMR 2638, Institut M. E. Chevreul, F-59000 Lille, France. [6] Univ. Lille, CNRS, UMR 8523, PhLAM, F-59000 Lille, France. ✉email: alessandro.faccinetto@univ-lille.fr

Although the central role of polycyclic aromatic hydrocarbons (PAHs) in the formation of nascent soot particles (NSPs) is now well established, a comprehensive understanding of the fundamental processes leading to the gas- to condensed-phase conversion (soot nucleation) in flame combustion has not been reached yet[1–4]. The hypothesis that soot formation might be kinetically rather than thermodynamically controlled and driven by the dimerization of small-to-moderate-size PAHs (C10–C18) dates back to the early 2000s[5,6]. However, the dimerization was initially considered as a convenient numerical tool to match the calculated and measured PAH concentrations and soot volume fraction profiles rather than a physically relevant phenomenon. In particular, thermodynamic calculations showing that only dimers of large PAHs would be sufficiently stable to survive at flame temperature[7] conflicted with the measured concentrations of large PAHs being too low to account for the observed number density of NSPs[8]. Early calculations of the entropic resistance to the binding of PAHs in equilibrium conditions showed that only dimers of PAHs as large as circumcoronene ($C_{54}H_{18}$) would be sufficiently stable to survive in a flame-like environment, and the formation of thermodynamically stable dimers was interpreted as a potential reversibility of the dimerization reaction[2]. Over the past few years, several papers exploring dimerization-based soot nucleation mechanisms appeared in the literature. Although the thermodynamic data of many PAHs are still subject to uncertainty, the reversibility of a nucleation process hypothetically driven by the dimerization of small PAHs has been used to successfully reproduce soot volume fraction profiles[9,10] and NSP size distributions[11] in several laboratory flames. To explain the dimerization of PAHs at high temperature, the formation of covalent carbon–carbon bonds between PAH monomers[12–18], or alternatively rapid radical-driven clustering reactions[19,20], has been invoked. The hydrogen abstraction acetylene addition (HACA) mechanism, the defining feature of which is the kinetic–thermodynamic coupling, has also been recently generalized by postulating the formation of doubly bonded covalent bridges between PAHs[18]. Although an exact mechanism has not yet been confirmed, all these works share the same conclusion that the dimerization of small-to-moderate-size PAHs is a viable process to explain the nucleation of NSPs. Nonetheless, experimental evidence of the formation of dimers of PAHs of any size in flames is still scarce[21–23]. The excellent predictive power of the models based on the reversibility of the dimerization clashes with the practical limits of in situ detection of trace amounts of short-lived species in flame conditions.

Data supporting the existence of dimers of small-to-moderate-size PAHs in flame combustion have been recently obtained from the analysis of visible-induced fluorescence spectra collected from a co-flow laminar diffusion methane–air flame that reveals two classes of fluorescing molecules attributed to PAH monomers and dimers discriminated by their spectroscopic signature[23]. This work addresses the need for further experimental data to confirm or disprove the existence of dimers of small-to-moderate-size PAHs in the flame and their connection, if any, to the soot-nucleation process. Information on the chemical composition of NSPs is obtained from the comparative analysis of soot and condensable gas extracted at different reaction times, equivalent to different heights above the burner (HAB), along the flame axis. The emission of broadband laser-induced incandescence (LII) identifies the soot region inside the flame[23], and it is used to select the key sampling points across the soot-formation region. The chemical composition of the samples is determined by ex situ time-of-flight secondary ion mass spectrometry (ToF-SIMS)[24–26], while information on amorphous carbon and nanocrystalline graphite and the size of the aromatic islands is obtained from

Raman spectroscopy[27]. Principal component analysis (PCA) is used to reduce the dimensionality of the database and find statistically significant data correlations[28,29].

## Results and discussion

**PAHs coexisting in the same mass range exhibit different behavior.** The results of the chemical analysis of the soot samples collected from the investigated diffusion flame are shown in Fig. 1. A detailed description of the flame and the sample preparation protocol is given in the section "Flame and sampling". Briefly, soot and condensable gas are extracted at five selected HABs through a dilutive microprobe and impacted on suitable substrates. As shown in Fig. 1a, at each HAB, two regions of interest (ROIs) are analyzed on the substrate surface by ToF-SIMS and Raman spectroscopy: the impaction site ROI and the surrounding halo ROI. Details are given in the corresponding "Methods" section. Then, PCA is performed on the ensemble of the collected mass spectra as discussed in the section "Principal component analysis". The output of the PCA is discussed in terms of scores (the coordinates in the principal component system) and loadings (the coefficients of the linear combinations used to build the principal components). Herein, a choice is made to limit the discussion and data interpretation to the first two principal components PC1 and PC2, as they explain by far the highest percentage of the total variance (89.27%). The PC2 vs. PC1 score plot is shown in Fig. 1b (each datapoint represents one mass spectrum), while the loading plots of PC1 and PC2 are shown in Fig. 1c and Fig. 1d, respectively. The interpretation of the score plot is based on two considerations. First, all acquisitions on the impaction ROI (solid symbols) are sorted in the score plot from the lowest to the highest HAB, increasing from the right to the left. Thus, PC1 (70.06% of the variance explained) discriminates the impaction ROI by the sampling HAB. The loadings of PC1 are shown in Fig. 1c. The negative loadings of PC1 feature a unimodal distribution characterized by the highest values in 226–276 $m/z$, and progressively decreasing at higher $m/z$. On the other hand, the positive loadings of PC1 feature a bimodal distribution characterized by a first mode at 152–252 $m/z$ and by a second mode at 326–440 $m/z$, roughly twice the $m/z$ of the first mode. Second, all acquisitions on the halo ROI (open symbols) have negative loadings in PC2, while all acquisitions on the impaction ROI except one (solid symbols) have positive loadings in PC2. Thus, PC2 (19.12% of the variance explained) discriminates different ROIs generated during the sampling by impaction deposition: soot particles mixed with condensable gas in the impaction ROI, and only condensable gas in the halo ROI. The absence of soot particles in the halo ROI is confirmed by the complete absence of Raman D (disordered structure of graphene) and G (stretching of C–C bonds in graphitic materials) bands. The loadings of PC2 are shown in Fig. 1d. The positive loadings of PC2 feature an irregular distribution that becomes homogeneous roughly above 300 $m/z$, while the negative loadings of PC2 are approximately constant up to 226 $m/z$ and then decay exponentially to zero up to 400 $m/z$.

The following discussion focuses on the role of the HAB on the interpretation of the data obtained from the impaction ROI. The score plot in Fig. 2 shows that the impaction ROI samples are well separated in three data clusters. Data cluster A (red dashed line) contains all mass spectra at 45 and 50 mm HAB, and it is dominated by PC1 > 0. No in-situ LII signal is detected at these HABs with the current experimental setup, and the Raman D and G bands have signal-to-noise ratio SNR < 3 in all the samples. Therefore, PC1 > 0 is regarded as characteristic of condensable gas upstream soot nucleation. Data cluster B (brown dashed line) contains only the mass spectra at 55 mm HAB, and it is

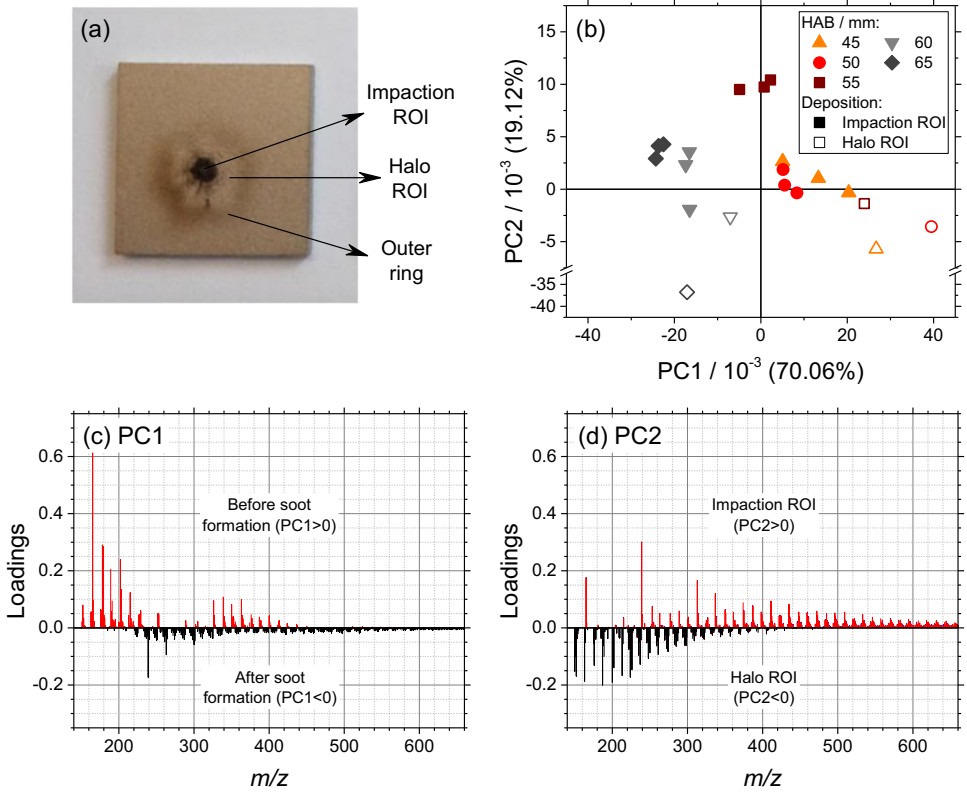

**Fig. 1 Results of PCA performed on 601 selected *m/z* in the ToF-SIMS mass spectra and representing the chemical composition of the soot samples.** Soot and condensable gas are impacted on Ti wafers for the ToF-SIMS analysis. **a** Definition of the ROIs. **b** PC2 vs. PC1 score plot. Each datapoint represents one mass spectrum. Color-coded symbols represent the sampling HAB, solid and open symbols indicate the selected ROI. **c, d** PC1 and PC2 loading plots, respectively. Red and black data represent the sign of the loadings.

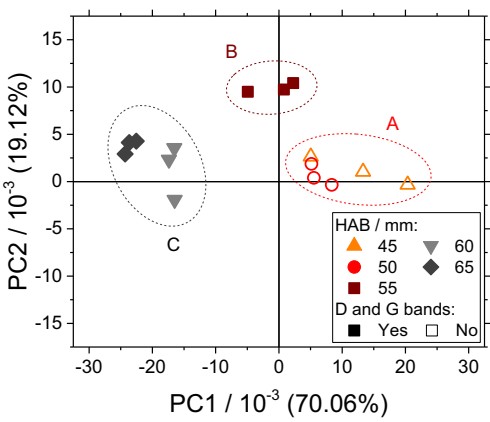

**Fig. 2 PC2 vs. PC1 score plot on the impaction ROI showing the three identified data clusters.** A (PC1 > 0, PC2 ≈ 0): Raman D and G bands have SNR < 3 and no LII is detected, is attributed to condensable gas. B (PC1 ≈ 0, PC2 > 0): Raman D and G bands have SNR ≥ 3 and LII has SNR < 3, is attributed to the transition region. C (PC1 < 0, PC2 ≈ 0): Raman D and G bands and LII all have SNR ≥ 3, is attributed to soot.

characterized by PC1 ≈ 0 and PC2 > 0. No LII signal with SNR ≥ 3 is detected yet in this transition region; however, the Raman D and G bands have risen above SNR ≥ 3. Regarding the cause of this discrepancy, searching the smallest incandescent particles by LII requires a specific configuration[30,31] that was not implemented in this work in favor of a configuration more suitable for mapping the flame, and therefore the existence of NSPs between 50 and 55 mm HAB cannot be completely ruled out. Data cluster

C (gray dashed line) contains all mass spectra at 60 and 65 mm HAB, and it is dominated by PC1 < 0. At these HABs, LII signal with SNR ≥ 3 is detected, and all samples feature intense Raman D and G bands. Therefore, PC1 < 0 is regarded as characteristic of soot downstream soot nucleation.

As shown in Figs. 1 and 2, several experimental findings converge to the conclusion that the chemical composition of the impaction ROI undergoes significant evolution through the transition region that is delimited upstream at 50 mm HAB (highest HAB identified in data cluster A) and downstream at 60 mm HAB (lowest HAB identified in data cluster C). The evolution of the mass spectra against the sampling HAB is represented by PC1 that explains 70.06% of the variance of the entire database. The contribution to the mass spectra of the *m/z* having positive loadings in PC1 (Fig. 1c) is much less important downstream than upstream the transition region. Remarkably, the positive loadings of PC1 feature a bimodal distribution that spans 152–252 *m/z* (mode 1) and 326–440 *m/z* (mode 2): while the disappearance of mode 1 downstream the transition region is well visible in the mass spectra (Supplementary Fig. 3), the disappearance of mode 2 is hidden and only revealed by the PCA. As previously mentioned, the transition region between 50 and 60 mm HAB (represented by data cluster B) is very important for this flame, as soot particles are detected for the first time within. The visual inspection of the samples (Supplementary Fig. 1) shows a change of the appearance of the sampled materials from yellow-brown and liquid-like at 50 mm HAB to black and dry at 60 mm HAB. In addition, at 55 mm HAB, Raman signal rises above SNR > 3 (Supplementary Fig. 4), and LII signal is detected for the first time[23]. In conclusion, this evidence suggests that the

disappearance of the $m/z$ having positive loading in PC1 from the condensable gas and the soot nucleation are correlated processes.

**The dimerization of small PAHs explains the bimodality of the positive loadings of PC1.** The observed bimodal distribution of positive loadings of PC1 characteristic of the condensable gas sampled in the flame axis upstream soot formation (impaction ROI, HAB ≤ 55 mm shown in Fig. 1c) is worthy of special attention. Specifically, we make the hypothesis that the bimodal distribution of loadings, which identifies strongly intercorrelated $m/z$ characteristic of the condensable gas sampled upstream soot formation, represents small hydrocarbons capable of dimerization (mode 1, 152–252 $m/z$) and their dimers (mode 2, 326–440 $m/z$). Accordingly, correspondences between pairs of $m/z$ in mode 1 and individual $m/z$ in mode 2 are looked upon as follows. First, unknown ions are identified by assigning a molecular formula ($C_mH_n^+$ and $C_mH_nO_p^+$) to the individual $m/z$ by mass defect analysis[24,26,32] as detailed in the section "ToF-SIMS". Then, correspondences are sought as $C_{m1}H_{n1}^+ + C_{m2}H_{n2}^+ = C_{m1+m2}H_{n1+n2-k}^+ + kH$, where $C_{m1}H_{n1}^+$ and $C_{m2}H_{n2}^+$ are ions in mode 1, $C_{m1+m2}H_{n1+n2-k}^+$ is the closest matching ion in mode 2, and $k$ represents the number of H atoms required to match formulae. In this analysis, only highly correlated $m/z$ and high confidence assignments are considered as detailed in the section "Analysis of the positive loadings of PC1". The identified correspondences are shown in Fig. 3a. Each row and column of the table represents one ion in mode 1 ($C_{m1}H_{n1}^+$ and $C_{m2}H_{n2}^+$). At the intersection of each row and column, i.e., for every pair of ions in mode 1, is a cell that contains the value of $k$ to match the closest ion in mode 2 ($C_{m1+m2}H_{n1+n2-k}^+$). An empty cell means that no correspondences are found. Isotopolog ions (not included in the table) show independent correspondences that further strengthen the confidence of the correspondence. For instance, for every $C_{m1}H_{n1}^+ + C_{m2}H_{n2}^+ = C_{m1+m2}H_{n1+n2}^+$, also $[^{13}C]C_{m1}H_{n1}^+ + C_{m2}H_{n2}^+ = C_{m1}H_{n1}^+ + [^{13}C]C_{m2}H_{n2}^+ = [^{13}C]C_{m1+m2}H_{n1+n2}^+$ are found. As shown in Fig. 3b, the number of correspondences against $k$ forms a distribution with a maximum at $k = 4$. The correspondences found for even $k$ values can be explained with the formation of C–C covalent bonds from pre-existing molecular species: 2H elimination is consistent with the hypothetical formation of one covalent bond; thus, $k = 2, 4, 6, 8$ would correspond to the formation of 1, 2, 3, 4 covalent bonds that in turn would lead to the formation of stable molecular structures more likely to be detected ex situ. Remarkably, no correspondences are found for odd $k$ values. The elimination of an even number of H possibly results in species too short-lived to be detected ex situ.

The possibility that the correspondences between pairs of ions in mode 1 with individual ions in mode 2 were artifacts due to the sampling or analytical protocol is accounted for and rejected. Theoretical arguments based on the collision probability support the idea that the formation of stable covalent bonds between hydrocarbons in the cold and diluted sample flow is unlikely. Similarly, the possibility of the dimerization being an artifact of the analytical protocol is excluded as tests on pure hydrocarbons show that the formation rate of dimers during the deposition and ToF-SIMS analysis is negligible. A detailed discussion is available in the section "The clustering of PAHs represents flame processes".

**Not all PAHs give rise to dimerization reactions.** The comparison of ToF-SIMS and Raman measurements provides useful information on the fraction of H atoms [H] of the impaction ROI. $[H]_{Raman}$ of hydrogenated carbon films deposited on conductive substrates can be obtained from Raman spectra through the

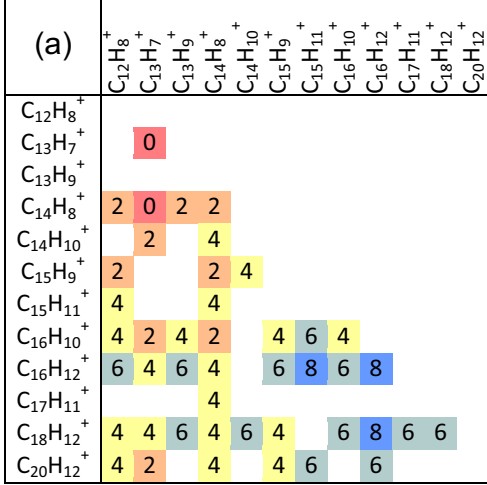

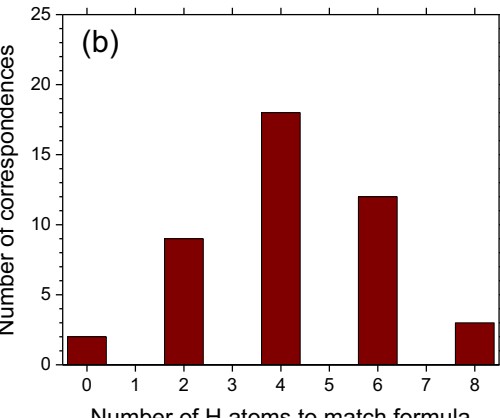

**Fig. 3 Identified correspondences between pairs of ions in mode 1 and individual ions in mode 2. a** Correspondences by molecular formula ($C_{m1}H_{n1}^+$ and $C_{m2}H_{n2}^+$ in mode 1 and $C_{m1+m2}H_{n1+n2-k}^+$ in mode 2). The table cells contain the values of $k$ required to match formulae, color-coded for easy visual interpretation. **b** Number of correspondences against number of H atoms to match formula $k$.

equation[33,34]

$$[H]_{Raman} = a + b \log \frac{m}{I(G)}, \tag{1}$$

where $m$ is the slope of the fluorescence, $I(G)$ the intensity of the G band, and a and b are empirical constants. From ToF-SIMS data, a simple yet original approach is developed to obtain the global fraction of H atoms $[H]_{ToF\text{-}SIMS}$, in which the normalized signal intensity of the mass spectra $w_i$ is used to weight the individual contributions of the identified ions:

$$[H]_{ToF-SIMS} = \frac{N_H}{N_H + N_C + N_O} \text{ with } N_X = \sum_i N_{X,i} w_i, \text{ X} = \text{H, C, O and } \sum_i w_i = 1, \tag{2}$$

In this approach, it is assumed that during ToF-SIMS analyses, the extraction and the ionization efficiency does not vary against $m/z$. As shown in Fig. 4, $[H]_{ToF\text{-}SIMS}$ increases monotonically against $\log m/I(G)$, and the increase in both $[H]_{ToF\text{-}SIMS}$ and $\log m/I(G)$ corresponds to the decrease in HAB. Although $[H]_{ToF\text{-}SIMS}$ and $\log m/I(G)$ are linearly correlated ($R^2 = 0.96$), the slope of the fitting function differs significantly from values obtained for hydrogenated carbon films (b = 0.013 to be compared with 0.166[33] and 0.046[34]). This discrepancy is

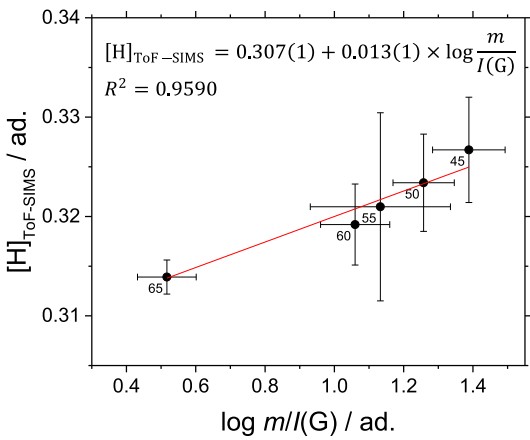

$$[H]_{ToF-SIMS} = 0.307(1) + 0.013(1) \times \log \frac{m}{I(G)}$$

$$R^2 = 0.9590$$

**Fig. 4 Calculation of the fraction of H atoms [H], [H]$_{ToF-SIMS}$ against log $m/I$(G) of the impaction ROI.** The error bars represent the standard deviation on all acquisitions and include the uncertainties on the Raman spectral data analysis.

attributed to the combination of two main factors. First, the range of log $m/I$(G) explored in this investigation is limited. Second, only the assigned molecular formulae are considered in the calculation of [H]$_{ToF-SIMS}$ that accounts on average for 78% of the total ion count of the analyzed samples. The remaining 22% of the total ion count corresponds to ions with low-mass defect (Supplementary Fig. 2) that could not be univocally identified due to the insufficient available resolving power, and potentially include carbon clusters and oxygen-containing hydrocarbons. Because of these unidentified ions, the linear fit shown in Fig. 4 can overestimate [H]$_{ToF-SIMS}$ by as much as 10% as detailed in the section "Equation (2) sensitivity analysis".

The size of the aromatic clusters $L_a$ is estimated from Raman spectroscopy measurements. For amorphous carbon and microcrystalline graphite, a parameter commonly used to estimate $L_a$ is the intensity ratio of the Raman lines $I(D)/I(G)$[27]. In all analyzed samples, $L_a$ is of the order of 1 nm, consistent with the size of PAHs having up to three–four fused aromatic rings per side. A clear trend against HAB could not be observed.

Molecular formulae are assigned to the highest positive loadings of mode 1 by crossing $m/z$ obtained from ToF-SIMS, [H] obtained from the comparison between ToF-SIMS and Raman spectroscopy, and $L_a$ derived from Raman spectroscopy. The molecular formulae are consistent with benzenoid PAHs ($C_{14}H_{10}^+$, $C_{16}H_{10}^+$, and $C_{18}H_{12}^+$), PAHs containing 5-member rings ($C_{12}H_8^+$ and $C_{16}H_{10}^+$), radical ions obtained from hydrogen elimination reactions from PAHs containing a five-member ring ($C_{13}H_9^+$, $C_{15}H_9^+$, and $C_{17}H_{11}^+$), and side-substituted PAHs ($C_{16}H_{12}^+$). These molecular formulae are also consistent with the structural formulae proposed in soot formation models based on the dimerization of low-mass PAHs[13,15–19]. The many correspondences between sums of pairs of $m/z$ in mode 1 with individual $m/z$ in mode 2 suggest that the dimerization of C12–C20 hydrocarbons having individual [H] 0.35–0.43 followed by the formation of C–C covalent bonds occurs in the flame axis immediately upstream soot formation. In particular, the identified ions confirm the participation of five-member ring PAHs as observed by atomic force microscopy[35] and predicted by theoretical modeling[16], worth of more in-depth investigations in future works. The comparatively small loadings of oxygen-containing hydrocarbons indicate that their contributions to PC1 have low statistical significance, and therefore their involvement in the dimerization process is deemed unlikely.

The formulae assigned to the highest loadings of mode 2 correspond to C26–C36 hydrocarbons having individual [H]

0.32–0.36 ($C_{26}H_{14}^+$, $C_{27}H_{15}^+$, $C_{28}H_{14}^+$, $C_{29}H_{15}^+$, $C_{30}H_{16}^+$, $C_{31}H_{15}^+$, $C_{32}H_{16}^+$, $C_{33}H_{17}^+$, $C_{34}H_{16}^+$, $C_{35}H_{17}^+$, and $C_{36}H_{18}^+$). For each pair of PAH monomers, dimers can initially form as van der Waals dimers or dimer radicals. However, since dispersion forces are too weak to overcome the entropy penalty of dimerization above 1000 K[7], van der Waals dimers and dimer radicals cannot survive at flame temperature, unless covalent C–C bonds form to stabilize them. Accordingly, many of the identified molecular formulae are consistent with the structural formulae expected for covalently stabilized homo- and heterodimers formed from monomers in mode 1[13,17], and with many structures observed by atomic force microscopy[14,35].

In conclusion, the experimental data presented in this work support the hypothesis of dimerization of small-to-moderate-size PAHs occurring in the condensable gas phase in the flame axis upstream soot formation. Principal component analysis discriminates two classes of correlated compounds, and the resulting bimodal distribution of loadings is explained with the dimerization of C12–C20 PAHs followed by the formation of C–C covalent bonds. The assigned molecular formulae are consistent with the PAH monomers and covalently stabilized dimers predicted by dimerization hypotheses, and might be directly linked to the nucleation of NSPs, despite the unfavorable thermodynamic conditions for their formation at flame temperature. In particular, the species identified as covalently stabilized dimers represent promising candidates to explain the broadband fluorescence observed in our previous work[23]. We expect the present work to be relevant for the development or improvement of kinetics modeling as it provides a much-demanded list of $m/z$ for both monomers and dimers to be used as input for the theoretical models.

## Methods
**Flame and sampling.** The investigated diffusion methane flame is stabilized at atmospheric pressure on a custom-built Gülder-type burner[24]. The central injector (12.7 mm outer diameter) is supplied with 0.52 L min⁻¹ of N55-grade methane, while the outer ring (60 mm outer diameter) is supplied with 86.6 L min⁻¹ filtered compressed air. These settings result in a 120 mm high nonsmoking flame. A windowed quartz chimney is installed on top of the burner to minimize flame perturbations. Pictures of the flame and the samples are shown in Supplementary Fig. 1. Five sampling points on the flame axis at different HAB are established based on the laser-induced fluorescence (LIF) map of the gas-phase precursors and the laser-induced incandescence (LII) map of the soot particles (already measured in our previous investigations[23]): at 45 and 50 mm HAB before the appearance of LII signal, at 55 mm HAB at the very beginning of the LII signal, and at 60 and 65 mm HAB in the positive slope side of the LII region. Soot and condensable gas are sampled using an extractive dilutive quartz microprobe and deposited by impaction on Ti wafers and quartz slabs for ToF-SIMS[25] and Raman spectroscopy[36], respectively. An extraction system consisting of a sampling microprobe coupled to an automatic pressure regulator is used to maintain a high dilution ratio (up to around 1000) during sampling, and thus reducing the probability of collisions in the sample flow while minimizing flame perturbations.

**ToF-SIMS.** ToF-SIMS analyses are performed at the Regional Platform for Surface Analysis of the University of Lille on a commercial ION-TOF GmbH ToF.SIMS[5] mass spectrometer equipped with a 25 kV, 3 pA Bi$_3^+$ primary ion gun. During ToF-SIMS analyses, samples are sputtered with a primary ion beam, and the ejected secondary ions form a plume expanding in the mass spectrometer ion source (residual $p \sim 10^{-8}$ mbar). The plume is sampled by an extraction cone, and the extracted ions are mass-analyzed with a time-of-flight mass spectrometer having maximum resolving power $m/\Delta m \sim 10^4$. The estimated ion dose of ~$10^{11}$ ions cm⁻² is well below the threshold of ToF-SIMS static mode. The lower detection limit is estimated at $10^7$ atoms cm⁻² and the depth sensitivity 1–2 nm. Mass spectra in positive polarity are recorded at 130 scans/acquisition (300 s) on a 500 × 500 µm² surface with an image resolution of 256 × 256 pixels. Three different zones are analyzed on each impaction and halo ROI. The collected mass spectra are aligned and calibrated. The identification of unknown peaks is based on the mass-defect analysis[32] on the ensemble of detected peaks. The mass defect plot is shown in Supplementary Fig. 2. The background and the fragment ions are identified and removed following the protocol detailed in our previous work[24], and based on the comparative analysis of different samples followed by hierarchical clustering analysis. Finally, the mass spectra are normalized by the total ion count after background and fragment ion removal. The resulting mass spectra, after background and fragment ion removal and normalization, are shown in Supplementary Fig. 3.

With respect to our previous investigations[25], an improved peak list containing 1020 peaks having SNR > 3 is used. In total, 419 peaks are identified as blank or fragment ions and removed. Of the 601 peaks attributed to soot and condensable gas phase and used for the PCA, 355 peaks (59%) are identified with high confidence and account on average to 78% of the total ion count. The remaining unidentified 246 peaks (41%), that account on average for 22% of the total ion count, are almost all located in the low-mass defect region of the mass defect plot. The confidence of the assignment of these signals is generally lower, and therefore it is not always possible to clearly distinguish oxygen-containing hydrocarbons from carbon clusters with the currently available resolving power. A detailed list of the ions used for the data reduction and interpretation is available in Supplementary Table 1.

**Raman spectroscopy**. The flame extracts are deposited by impaction (around 30 m s$^{-1}$ impaction velocity) on fused silica slabs for the Raman analysis. The slabs, without any further manipulation, are positioned under the Raman microscope (Horiba XploRA) equipped with a 100× objective (NA 0.9, Olympus), corresponding to a laser spot of about 1–2 μm. The laser source is a frequency-doubled Nd:YAG laser ($\lambda = 532$ nm, 12 mW maximum laser power). The power of the excitation laser beam and the exposure time is adjusted to avoid structural changes of the sample due to thermal decomposition. Spectra are obtained with a laser beam power of 1%, about 0.1 mW on the sample, and an accumulation–exposure time of five cycles of 20 s each. The calibration of the system is performed against the Stokes Raman signal of pure silicon at 520 cm$^{-1}$ and the G line of highly ordered pyrolytic graphite line at 1581 cm$^{-1}$. A 200 μm pinhole is used for confocal photon collection. For each slab, measurements are done focusing the microscope on several points on the impaction ROI and on the halo ROI. The Raman spectra in the range of 800–2200 cm$^{-1}$ of the soot samples collected at different HAB are shown in Supplementary Fig. 4. The red rectangles highlight the D and G Raman bands. By increasing the HAB, the SNR sensibly decreases. The intensity of the D and G bands increases with respect to the background, with increasing HAB. The relative intensity of the D and G bands, $I(D)/I(G)$, is dependent on the size of the graphitic domains $L_a$ (nm)[27]. For the very small size of the graphite crystallites, as in the case of the soot samples herein investigated, the following expression is used to correlate $L_a$ with the relative intensity of the two Raman peaks[27]:

$$L_a^2(\mathrm{nm}^2) = 5.4 \times 10^{-2} E_L^4 \frac{I(D)}{I(G)}, \qquad (3)$$

where $E_L$ is the energy of the incident photon (eV). The uncertainty in the $L_a$ is estimated of the order of 20% based on the standard deviation of the data and the uncertainties on the spectral data analysis for the evaluation of the peak intensities[27,33,36].

**Principal component analysis**. PCA operates a series of linear transformations of the original variables to find a new set of orthogonal variables in the directions that maximize the variance of the analyzed population. These new variables, called principal components, are sorted in descending order on the percentage of the variance explained (PC1, PC2, and so on). In this work, the PCA is performed on the covariance matrix using individual $m/z$ as variables and acquisitions on the impaction ROI and halo ROI at each HAB as observations. The percentage of the variance explained by PC1 (70.06%, effect of HAB) is much larger than PC2 (19.12%, effect of ROI). In all, 10.82% of the variance remains unexplained. When using PCA, it is important to be aware that the loading plots do not show the variable intensity, but rather the coefficients of the linear combinations used to build the principal components. In PCA, high loadings (regardless of the sign) indicate that the corresponding variables have a strong relationship with their principal component. Loadings close to zero indicate that the variable does not affect the phenomenon described by the principal component.

**Analysis of the positive loadings of PC1**. The correspondences between couples of $m/z$ in mode 1 and individual $m/z$ in mode 2 are identified by applying two different criteria. First, all the ions have to be identified with high confidence. In high-resolution mass spectrometry, the relative difference $\delta = \mathrm{abs}(m/z - M_r)/m/z$, where $m/z$ is the accurate mass and $M_r$ is the exact mass of the assigned hydrocarbon, is used to estimate the confidence given to assignments. $\delta \leq 5$ ppm is often considered to be a certain assignment. $\delta \leq 15$ ppm is considered as acceptable with the currently available resolving power because of the few elements in the samples (only H, C, and O) that result in a small number of different molecular formulae consistent with a given $m/z$. Figure 5a shows the number of correspondences found against $k$ for different $\delta$ up to 50 ppm. The general trend does not change with $\delta$, and this is considered as a test of the robustness of this selection approach. Second, the ion in mode 2 has to be highly correlated to the two ions in mode 1. Figure 5b shows a zoom on mode 2 of the loading plot of PC1. Only the $m/z$ having the highest loading of each group is selected, while all satellite $m/z$ are discarded.

**The clustering of PAHs represents flame processes**. The possibility of dimerization occurring inside the probe during sampling, during the impaction deposition on the wafer surface or in the gas plume in the ion source of the mass spectrometer during the analysis, has been considered and is discussed in this

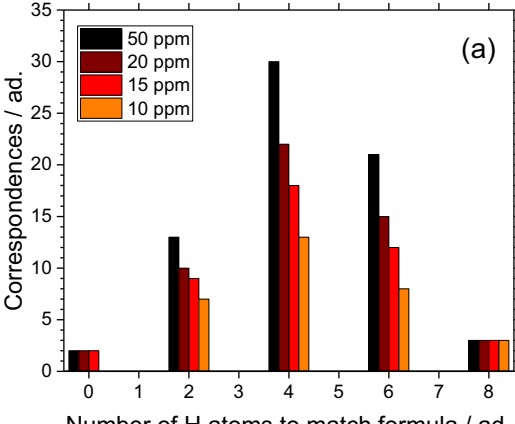

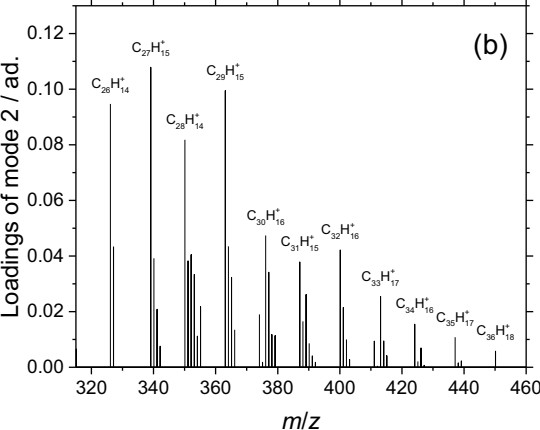

**Fig. 5 Criteria used for identifying the correspondences between pairs of ions in mode 1 and individual ions in mode 2. a** Number of correspondences found against $k$ for different $\delta$. **b** Zoom of the loading plot of PC1 on mode 2. Only the labeled $m/z$ (the highest loading of each group) are selected.

section. From the collision theory, the number of collisions per unit volume and per unit time $Z$ is given by

$$Z = n_A n_B \sigma_{AB} \overline{v_T}, \overline{v_T} = \sqrt{\frac{8 k_B T}{\pi \mu_{AB}}}, \qquad (4)$$

where $n_A$ and $n_B$ are the number concentrations of the two molecules, $\sigma_{AB}$ is the collisional cross section, and the mean relative thermal velocity $\overline{v_T}$ is derived from the Maxwell–Boltzmann distribution as a function of the temperature $T$ and the reduced mass $\mu_{AB}$. $k_B$ is Boltzmann's constant. For identical molecules, the equation above reduces to

$$Z = \frac{n_A(n_A - 1)}{2} \sigma_A \overline{v_T} \approx \frac{1}{2} n_A^2 \sigma_A \overline{v_T}, \overline{v_T} = \sqrt{2}\sqrt{\frac{8 k_B T}{\pi m_A}}, \qquad (5)$$

where the additional 1/2 ensures that the collisions are not counted twice. To simplify the calculations, instead of the 36 PAHs identified having negative loading in PC1, only one molecule having mass roughly in the center of the distribution (pyrene, $m = 3.36 \times 10^{-23}$ kg) and concentration equal to the estimated total concentration of all PAHs ($x_{tot} = 3.6$ ppm) is considered. For an estimated flame temperature of 1600 K and using the ideal gas law, $n_{PAH} = 1.65 \times 10^{19}$ m$^{-3}$ of reactive PAHs in the flame. After sampling, due to the temperature drop and dilution (estimated dilution factor 500–1000), $n_{PAH} = 9.01 \times 10^{16}$ m$^{-3}$. The estimated mean velocity $\overline{v_T}$ is 409 m s$^{-1}$ in the flame and 175 m s$^{-1}$ in the sampling line. These data lead to $Z_{flame} = 1.10 \times 10^{22}$ m$^{-3}$ s$^{-1}$ and $Z_{line} = 1.40 \times 10^{17}$ m$^{-3}$ s$^{-1}$, roughly five orders of magnitude smaller in the sampling line than in the flame that clearly shows how the sampling greatly reduces the probability of collision and the gas temperature at the same time, thus making formation of covalent bonds post sampling unlikely.

Additional experiments have been performed in order to estimate the probability of molecular clustering of low- and moderate-size PAHs in the plume

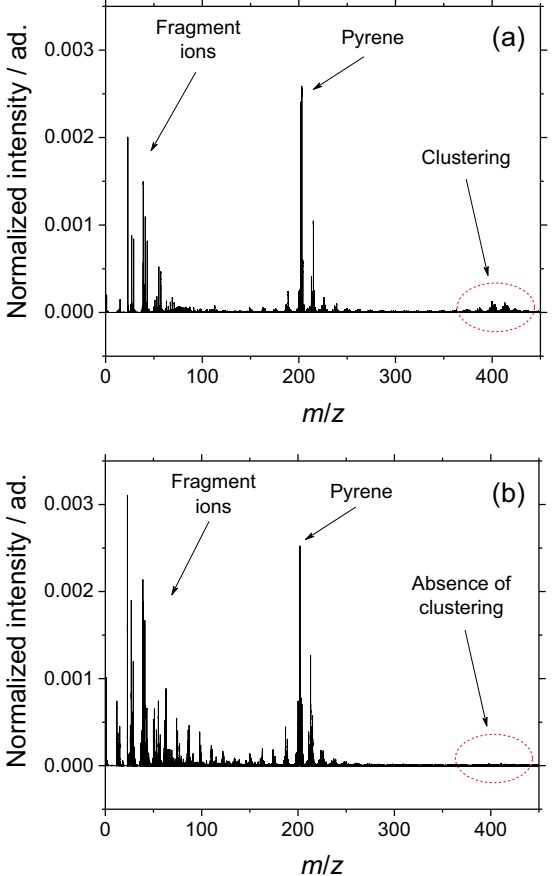

**Fig. 6 Clustering of pyrene during ToF-SIMS analyses.** Mass spectra of (**a**) pyrene, $c_{py} = 6.03$ M and (**b**) pyrene adsorbed on black carbon $c_{py} = 1.00 \times 10^{-2}$ M to simulate soot. Several fragment ions can be identified on the left side of the mass spectra. The weak signals at $m/z$ 400–420 in panel (**a**) are attributed to clustering reactions.

of desorbed gas during ToF-SIMS analyses. In particular, two cases have been examined:

(a) Pure PAHs. Pyrene, fluoranthene, and coronene (≥96% purity purchased from Sigma Aldrich) have been deposited on Ti wafers (one droplet of a concentrated solution in dichloromethane has been placed on the wafer surface; then the solvent evaporated with a gentle flow of clean nitrogen), then analyzed by ToF-SIMS (estimated bulk concentrations on the wafer surface: $c_{py} = 6.03$ M, $c_{fl} = 5.93$ M, $c_{co} = 4.38$ M). The mass spectrum of pyrene in Fig. 6a is shown as an example: at such high concentration, some clustering occurs during ToF-SIMS analyses, most likely as a result of post-ionization processes in the gas plume. For all investigated PAHs, the signals of molecular clusters are weak (2.0–2.5% of the PAH signal).

(b) PAHs adsorbed on black carbon to simulate the surface of soot particles. PAHs adsorbed on black carbon have been used in our previous works as standards to characterize sensitivity and detection limit of the two-step laser-desorption ionization (L2MS) experiments[37]. The ToF-SIMS mass spectrum of pyrene (bulk concentration $c_{py} = 1.00 \times 10^{-2}$ M) of one such sample is shown in Fig. 6b. The signals of molecular clusters are absent.

During the analysis of soot samples extracted from the diffusion flame investigated in this work, the accurate measurement of the bulk concentration of individual PAHs deposited on the wafer surface was not possible. However, estimations based on the known flame concentration of PAHs, sampling flow and time, and deposited spot size yield the upper limit $c = 2 \times 10^{-5}$ M. This value is more than five orders of magnitude smaller than the concentration of the tested pure PAHs, and three orders of magnitude smaller than the concentration of the tested PAHs adsorbed on black carbon, even assuming 100% deposition efficiency. The weak signal of a molecular cluster obtained with high concentration PAHs, when compared with the strong signals from flame extracts originating from much lower PAH concentrations, is therefore considered as clear evidence that clustering reactions during the ToF-SIMS analysis, although possible, occur at negligible rate when compared with flame processes.

**Equation 2 sensitivity analysis**. The uncertainties shown in Fig. 4 represent the random errors calculated from the standard deviation of three independent measurements, and then propagated using an approximation to a first-order Taylor series expansion. As briefly mentioned in the main text, the systematic error introduced by the existence in the peak list of unknown species ($m/z$ used for the PCA but not identified) can be difficult to estimate since, if a molecular formula is not known, it cannot be included in the calculation of $[H]_{ToF\text{-}SIMS}$ with Eq. (2). The partial ion count of unknown ions is around 22% of the total ion count. Although it might be tempting using this value to estimate the confidence given to $[H]_{ToF\text{-}SIMS}$, if some speculation is accepted, a better estimation can be based on the mass defect analysis as it follows. In the mass defect plot shown in Supplementary Fig. 2, unknown ions are mostly located in region (4) below the main sequence of PAHs. Their mass defect $\Delta$ is positive but close to zero, even at high $m/z$. Since H, C, and O are by far the main contributors to the chemical composition of the analyzed samples, two scenarios are consistent with the experimentally observed $\Delta \approx 0$:

(a) H-poor carbon clusters: $\Delta(C) = 0$ and $\Delta(H) > 0$, so molecular clusters or fragment ions containing a high number of C and up to 2–3 H would result in $\Delta$ close to the experimentally observed values.

(b) H-rich/O-rich clusters: $\Delta(H) > 0$ compensates for $\Delta(O) < 0$. In addition, since $\Delta(H) = +0.0078$ and $\Delta(O) = -0.0051$, to keep $\Delta \approx 0$, the ratio between H and O should be roughly constant at 2H:3O.

To estimate the systematic error, a molecular formula is tentatively assigned to the remaining unknown ions according to scenarios (a) and (b); then $[H]_{ToF\text{-}SIMS}$ is recalculated and compared with Eq. (2). Keeping in mind Eq. (2), the overall low $N_O$ in the collected soot samples also imposes a low $N_H$ in scenario (b), and is similar to scenario (a). Therefore, in practice, both scenarios yield $[H]_{ToF\text{-}SIMS}$ values ~ 10% lower than the value calculated by neglecting unknown ions (shown in Fig. 4).

## Data availability
The data that support the findings of this study are available from the corresponding author upon reasonable request.

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

## Acknowledgements

This work was supported by the Agence Nationale de la Recherche through the LABEX CAPPA (ANR-11-LABX-0005), the Ministry of Higher Education and Research, Hauts de France Regional Council, and European Regional Development Fund (ERDF) through the Contrat de Projets Etat Region (CPER CLIMIBIO), the H2020 project Portable Nano Particle Emission Measurement System (PEMs4Nano, Grant Agreement no. 724145).

## Author contributions

Each named author has substantially contributed to the research. A.F., C.I., P.M., C.P., and X.M. conceived and planned the experiments. A.F. and X.M. prepared the samples. A.F., C.I., and N.N. performed the ToF-SIMS analysis, data reduction, and interpretation. P.M., M.C., and C.P. performed the Raman analysis, data reduction, and interpretation. A.F., C.I., P.M., M.C., and C.P. developed the comparison method between ToF-SIMS and Raman analyses. A.F., C.I., and P.M. wrote the paper with contributions from all authors. A.D.A., Y.C., P.D., C.F., and X.M. provided the funding, contributed to the data interpretation and the discussion, and were in charge of overall direction and planning.

## Competing interests

The authors declare no competing interests.
