## [Peer Review File · Communications Chemistry]

Reviewers' comments:

Reviewer #1 (Remarks to the Author):

Review for COMMSCHEM-20-0097

This article presents valuable and enlightening results on evolution of chemical composition of species during soot formation that help us better understand nucleation of soot and other carbonaceous nanoparticles (such as interstellar dust). This work has three main novelties:

1. It applies Principal Component Analysis to reveal hidden trends in ToF-SIMS mass spectra for the first time in the combustion community (to the best of my knowledge).
2. It combines three different (and somehow orthogonal) measurement methods to follow the evolution of the chemical composition of species. Simultaneous use of these methods together with PCA enables the authors to link trends in ToF-SIMS mass spectra to soot particles and their precursors.
3. It illustrates for the first time that dimers of PAHs with weak Vander Waal forces may form first and then stabilized by formation of carbon-carbon covalent bonds.

I think the paper deserves publication in Nature communications. However, The authors can improve the paper so that it is easier to understand it.

Here are some questions and comments:

1. Figure 1d, both top and bottom of the figure show (PC2<0), I think the top one should be (PC2>0)
2. It is mentioned that :
"The change of sign of PC1 at 55 mm HAB corresponds to the beginning of the LII signal and for this reason is considered as a marker of the formation of (incandescent) soot particles"
and
"This clustering represents a change in the chemical composition of the condensable gas between 55 and 60 mm HAB although a detailed explanation is not yet available, the detection of a weak LII signal at 55 mm HAB suggests that the disappearance of compounds below 400 m/z from the condensable gas and the soot nucleation are linked processes"

I am not convinced that change of sign for PC1 around HAB = 55 mm signals change in chemical composition. The origin of PC1 is centered at the average of the data. If I understand it correctly, at 55 mm, the score of positive and negative loadings for PC1 are equal. The fact that at HAB = 55 mm we start to get Raman signal and low quality LII signal may be the only evidence of change in the composition.

3. The discussion about Figure 2 on the last paragraph of page 6 and the first paragraph of page 7 is very important, my question is, which of these techniques, LII and Raman are more sensitive to changes in the chemical composition of the sample?

4. What do the authors mean by "At the crossing of every row and column, i.e. for every pair of ions in mode 1, the mismatch (ppm) to the closest m/z in mode 2 is given. The distances are color-coded for easy visual interpretation, green for certain matches (< 5 ppm) down to red for very unlikely matches (> 50 ppm).", to be more specific, my question is what role concentration plays here? are concentration numbers the confidence intervals?

5. There are no matches found for "correspondences -1H ", could it be because such structures are very short lived? Because to form -2H matches from direct match there is large energy barrier for simultaneous loss (abstraction) of two hydrogen atoms.

6. In table 1, for "direct match" mostly ions with up to 16 carbon atoms are found, however, the map for -2H matches is wider and even ions with 20 carbon atoms are found. Why is that? Could it be because direct matches for larger ions have lower concentration but covalently bounded matches are more stable and more likely to be found? Is it possible to comment on the relative concentration of direct matches and those with -2H matches?

7. "355 peaks (59%) are identified with high confidence (< 5 ppm) ", is 5 ppm the uncertainty in the concentrations?

8. I do not understand the logic behind Figure 3. Also, I have difficulty understanding how estimating "La" helps us reduce possible contributors to soot as all combinations are below the 1 nm threshold.

Reviewer #2 (Remarks to the Author):

The paper addresses an area of significant importance with both scientific and practical implications and as such is likely to be of interest to readers. In the main the paper is reasonably well written with some interesting findings, e.g. the confirmation of the absence of soot particles in the condensable "halo" material and the approximate bimodality of the in m/z prior to soot formation. The discussion regarding Fig. 2 is also excellent, clear and so on. However, some revision is required prior to publication with main issues below:

1. The discussion on thermodynamically-controlled soot formation makes two omissions that must be corrected. Firstly, the work by the Stanford group of Wang (e.g. Wang, Formation of nascent soot and other condensed-phase materials in flames, Proc. Combust. Inst. 33(1) (2011) 41-67) suggests entropy as a driving force through dehydrogenation reactions. Secondly, there are significant uncertainties in the determination of thermodynamic data for large PAHs as discussed by the Imperial College group (e.g. Lindstedt and Waldheim, Modeling of soot particle size distributions in premixed stagnation flow flames, Proc. Combust. Inst. 34(1) (2013) 1861-1868) that needs to be recognised in the context of reversibility. The latter work also shows that soot PSDs can be computed accurately starting with dimerization, while also recognising the need for work of the current kind due to the necessary transition from PAH dimerization to the van der Waals enhanced collision regime.

2. The role of radical reactions is also highlighted in the work mentioned under point 1 and the importance of C5 ring structures in PAHs is relevant in this context as, for example, some resonantly stabilised radicals (e.g. indenyl) become prevalent in flames. (e.g. Lindstedt, Chemical Complexities of Flames, Proc. Combust. Inst. 27 (1998) 269-285). Also see the related discussion on p. 11-12 in the current paper. As stated, there has been recent further corroboration of such observations. However, the current paper should be amended in this regard to recognise that the issue was established quite some time ago.

3. The use of abbreviations is very prevalent. This is fine. However, PC2 and PC1 are introduced in a haphazard way on p. 4 with essential information relegated to Fig. 1 (e.g. what PC1 > 0 stands for). The authors should correct such matters to make the paper more readable. The data presented is, however, interesting.

4. There appears to be an error in Fig. 1(d) where both the Impaction ROI and Halo ROI are defined by PC2 < 0. Certainly, the text suggests a different interpretation or perhaps the writing is in need of attention. The 19% PC2 result is poor and needs to be commented on further. By

contrast, the PC1 value of 70% is quite impressive.

5. In terms of the Halo and Impaction ROIs, the paper fails to make reference to the earlier work of Abid et al. (e.g. Abid, Heinz, Tolmachoff, Phares, Campbell, Wang, Combust. Flame 154 (2008) 775–788 and Abid, Tolmachoff, Phares, Wang, Liu, Laskin, Proc. Combust. Inst. 32 (2009) 681–688). This should also be corrected.

6. The top of p. 6 reads "...clusters. 45, 50 and 55 mm HAB.." This is not an acceptable way of writing. Please correct. The valuable technical conclusion that there is a change in the chemical composition downstream in the flame is credible, but also obscured by the writing.

7. On p. 8 it is argued that artifacts due to sampling and analysis are insignificant. This is a key point in the paper and the reference should be made correctly to the methods section.

8. The global fraction of [H] content is very interesting and well-performed. While the R2 norm is excellent, there are large uncertainties and it is not stated why (the very credible observation) that -O containing species could be a cause and, if so, to what an extent. It is accepted that this is speculation, but it should be more firmly grounded.

9. The discussion on p. 12 concerning vdW forces should usefully also refer to point 1 above.

The subsequent discussion is fine and the methods discussion shows the commendable care taken by the authors. The expectation is that the above revisions can readily implemented and that this would result in a paper suitable for publication.

Reviewer #3 (Remarks to the Author):

The paper studies the dimerization of PAH which is thought to be the source of soot nucleation but much is unknown. The possibility of dimers of low mass PAH resulting in soot nucleation has been much debated but to date there has not been experimental confirmation. The paper provides a novel and convincing argument supported by detailed experiments that dimers of low-mass PAH occur just before soot nucleation and supports the reactive dimerization model. The paper provides a detailed discussion of the chance that the dimers have been formed in the probe, on the wafer or on soot particles rather than in the gas phase. This paper marks significant progress in our understanding of soot nucleation and will be highly influential to the field.

Detailed Comments:

On pages 7 and 18, you say that stable covalent bonds cannot be formed during sampling; but could dimers formed by Vander Waals forces be possible in the sampling system?

Do all data points in figure 1c correspond to $PC2 > 0$? If so, you should clarify.

Why do both the halo and impaction zone of PC2 (figure 1d) correspond to $PC2 < 0$? This seems to contradict the statement at the bottom of page 4,

11th June 2020

**Object – Answers to the Reviewers for the paper: “Evidence on the formation of dimers of polycyclic aromatic hydrocarbons in a laminar diffusion flame”
Ref. COMMSCHEM-20-0097**

Dear Reviewers,

We would like to thank you for the amount of work that you clearly dedicated to our paper “*Evidence on the formation of dimers of polycyclic aromatic hydrocarbons in a laminar diffusion flame*”, we greatly appreciated your constructive comments and suggestions. We have carefully studied all comments and thoroughly revised the original paper accordingly. Please see the detailed point-by-point answers at the end of this document, and the attached revised paper “Revised manuscript COMMSCHEM-20-0097 final marked up.pdf”. In particular:

- Some comments prompted an expanded data analysis that is now detailed in the Results section.
- In all the sections where a lack of detail was noticed or clarifications requested, we expanded and completed the discussion.
- As suggested, the terminology has been made more consistent.
- Although not explicitly demanded, the text has been slightly modified in several places to include in the discussion the very pertinent Frenklach M. and Mebel A. M., Phys. Chem. Chem. Phys. 22 (2020) 5314-5331 that was published during the review process of our paper.

With our best regards,

Dr. Alessandro Faccineto on behalf of all Authors.

Reviewer #1

This article presents valuable and enlightening results on evolution of chemical composition of species during soot formation that help us better understand nucleation of soot and other carbonaceous nanoparticles (such as interstellar dust). This work has three main novelties:

1. It applies Principal Component Analysis to reveal hidden trends in ToF-SIMS mass spectra for the first time in the combustion community (to the best of my knowledge).
2. It combines three different (and somehow orthogonal) measurement methods to follow the evolution of the chemical composition of species. Simultaneous use of these methods together with PCA enables the authors to link trends in ToF-SIMS mass spectra to soot particles and their precursors.
3. It illustrates for the first time that dimers of PAHs with weak Vander Waal forces may form first and then stabilized by formation of carbon-carbon covalent bonds.

I think the paper deserves publication in Nature communications. However, the authors can improve the paper so that it is easier to understand it. Here are some questions and comments.

Comment 1. Figure 1d, both top and bottom of the figure show ($PC2 < 0$), I think the top one should be ($PC2 > 0$).

Answer. The impaction ROI is supposed to correspond to $PC2 > 0$. This error in Figure 1d has been corrected.

Comment 2. It is mentioned that : "The change of sign of PC1 at 55 mm HAB corresponds to the beginning of the LII signal and for this reason is considered as a marker of the formation of (incandescent) soot particles" and "This clustering represents a change in the chemical composition of the condensable gas between 55 and 60 mm HAB although a detailed explanation is not yet available, the detection of a weak LII signal at 55 mm HAB suggests that the disappearance of compounds below 400 m/z from the condensable gas and the soot nucleation are linked processes". I am not convinced that change of sign for PC1 around HAB = 55 mm signals change in chemical composition. The origin of PC1 is centered at the average of the data. If I understand it correctly, at 55 mm, the score of positive and negative loadings for PC1 are equal. The fact that at HAB = 55 mm we start to get Raman signal and low quality LII signal may be the only evidence of change in the composition.

Answer. We understand the concern of the Reviewer, however the information obtained from LII and Raman is used to support and understand the results of the PCA on the mass spectra that represent the change in the chemical composition of the samples. The individual mass spectra are normalized by the total ion count before building the covariance matrix for the PCA. This initial step in the data processing guarantees that the resulting principal components are representative of the variance within the dataset. The origin of PC1 was not centered at the average of the data. Since the scores are the coordinates of the datapoints in the new principal component space, the scores of 55 mm HAB ROIs near zero in PC1 simply means that these points are close to the average. The above mentioned change in the chemical composition of the samples refers to the evolution of the mass spectra between 50 and 60 mm HAB. The information that PCA extracts from the mass spectra database is the change in the relative intensity of the hydrocarbon signals in the impaction ROI occurring between 50 mm HAB (last scores of the HAB found on the PC1 positive axis) and 60 mm HAB (first scores of the HAB found in PC1 negative axis). This change is also visible in the normalized mass spectra (please see Supplementary Figure 3), and consists in the disappearance of many low-mass hydrocarbons (up to approximately

200-250 m/z) at 60-65 mm HAB with respect to 45-50 mm HAB (mode 1). The intriguing information that prompted the writing of this paper is the existence of a second “hidden” mode of positive loadings (mode 2) located at roughly twice the m/z of mode 1 and that could not be observed with the standard methods used in the analysis of mass spectra. The samples collected between 50 and 60 mm HAB belong to a region of interest for this flame. A simple visual inspection of the samples (Supplementary Figure 1) shows a transition of the material deposited on the wafers from yellow-brown and liquid-like at 50 mm HAB to black and dry at 60 mm HAB. The Raman signal rises above $SNR > 3$ (Supplementary Figure 4). Low LII signal is first detected below 60 mm HAB in the centerline of the flame (Irimiea et al., Carbon 144 (2019) 815-830; Mercier et al., Phys. Chem. Chem. Phys. 21 (2019) 8282-8294). Weak LII signal with $SNR < 3$ is observed from 50-55 mm meaning that incandescence from NSP is detectable from this HAB. The weakness of the LII signal (proportional to the number and the volume of the particles) at this HAB comes from the fact that at the very beginning of the soot formation process the particles are very small (a few nm) and scarce. Although not as clear as for the impaction ROI, the separation 45-50 mm HAB vs. 60-65 mm HAB is also preserved in the halo ROI as shown in Figure 1b. We acknowledge that this could have been better explained, and the text has been corrected and expanded accordingly.

Comment 3. The discussion about Figure 2 on the last paragraph of page 6 and the first paragraph of page 7 is very important, my question is, which of these techniques, LII and Raman are more sensitive to changes in the chemical composition of the sample?

Answer. In this work, the chemical composition of soot particles is obtained from the ToF-SIMS mass spectra, while LII and Raman are meant to provide, as the Reviewer correctly stated, “orthogonal” information for the interpretation of the PCA. The mass spectra are obtained with a commercial IONTOF⁵ apparatus that can achieve depth sensitivity of around 1 nm and a detection limit as low as 10^7 atoms cm^{-2} that makes it a very suitable choice for the analysis of ultratraces (this information has been added in the Methods section).

LII provides information on the soot volume fraction, and some implementations specifically developed for the detection of NSPs in premixed nucleation flames can reach a detection limit in the ppt range or even lower (Desgroux et al., Combust. Flame 184 (2017) 153-166). However, the configuration used in this work is developed for the full-scale characterization of the diffusion flame rather than searching for NSPs. Therefore, the LII signal from NSPs, especially when they are present in small quantities in the sampled volume, can fall in the noise threshold. The working principle of LII is based on the detection of the Planck radiation emitted by the soot particles, which are considered “gray bodies”, and thus only gives indirect information about the absorption properties of soot particles through the soot refractive index absorption/emission function. The LII signal depends on several physico-chemical properties of the particles, among which their surface chemical composition.

Raman spectroscopy is sensitive to the presence of fluorescing species, to C-C bonds in graphitic materials and to the number concentration of defects in the graphitic-like structures. The PAH signatures are relatively weak when using “conventional” Raman (Shinohara et al., J. Mol. Struct. 442 (1998) 221-234). Meanwhile, in visible Raman such as used here, the fluorescence background increases with the H content (Casiraghi et al., Diam. Relat. Mater. 442 (2005) 1098-1102) and sp^2 clustering. Therefore, any chemical characterization through the identification of specific Raman bands of individual PAHs rapidly becomes impossible. In contrast, structural information can be readily derived from Raman spectra using the D and G bands, provided that the fluorescence background remains moderate. In this work, the spatial resolution of the micro-Raman used is about 1 μm . On both the impaction ROI and the halo ROI of each sample, measurements are done focusing the microscope on several points obtaining the average spectrum. The differences in the

recorded spectra are about 10% and are attributed to lack of homogeneity of the soot deposits.

A direct comparison between Raman and LII is not trivial since the two techniques operate on a very different sampling logic. It is essential to remember that LII is an in-situ technique while Raman and ToF-SIMS are ex-situ techniques, therefore any comparison would be biased by the sampling procedure. The main advantage of LII is that it preserves (most of) the physical properties of the aerosol during the measurement, while Raman and ToF-SIMS can only provide information on the stable compounds that survive the sampling.

Comment 4. What do the authors mean by "At the crossing of every row and column, i.e. for every pair of ions in mode 1, the mismatch (ppm) to the closest m/z in mode 2 is given. The distances are color-coded for easy visual interpretation, green for certain matches (< 5 ppm) down to red for very unlikely matches (> 50 ppm).", to be more specific, my question is what role concentration plays here? are concentration numbers the confidence intervals?

Answer. Being the answer to this comment strictly related to the answers to Reviewer1, comments (5) and (6) and Reviewer3, comment (1) one comprehensive answer is given here. Briefly, these four comments, and in particular the request of Reviewer1 to discuss the role of the relative concentration of the different correspondences, prompted a more detailed analysis of the bimodal distribution observed in PC1>0.

In our original analysis, to identify the correspondences between couples of m/z in mode 1 and individual m/z in mode 2 we only used one condition on the mass relative uncertainty $\delta = \text{abs}(m/z - M_r)/m/z$ where M_r is the (assigned) exact mass. In high resolution mass spectrometry, δ (given in ppm) is often used to estimate the confidence given to assignments. $\delta \leq 5$ ppm is often considered to be a certain assignment. $\delta \leq 20$ ppm is considered as acceptable with the currently available resolving power because of the few elements in the samples (only H, C and O) that results in a small number of different molecular formulae with close M_r . $\delta > 20$ ppm should be "taken cautiously" because such large values can be caused by the coexistence of unresolved peaks at very close m/z or even by wrong assignments. This said, the idea behind Table 1 was to show that, for some couples of *identified* low-mass PAHs *classified* by PCA in mode 1, there are m/z in mode 2 that correspond to the sum of their individual m/z within the confidence interval mentioned above (green values). On the other hand, for other couples of PAHs *classified* by PCA in mode 1, the closest m/z in mode 2 is too far away from the sum of m/z in mode 1 to be considered a potential match, or sometimes even a PAH (red values). We acknowledge that all this could have been better explained, and we thank the Reviewer for this remark.

This said, in this revised version of the paper a more detailed analysis is developed. For each correspondence $C_{m1}H_{n1}^+ + C_{m2}H_{n2}^+ = C_{m1+m2}H_{n1+n2-k}^+ + kH$, the value k represents the number of H atoms to be removed from the sum of m/z in mode 1 to match the closest m/z in mode 2. Here, we extended the explored k-range from [0, 2], used in the original data processing protocol, up to [0, 9]. Furthermore, we added one selection condition on the loadings as they essentially represent the statistical significance of the correlation between signals. Essentially, we chose to focus on the main m/z of each group and to discard all satellite m/z as shown in the loading plot below (figure 1, zoom on mode 2 in the loading plot of PC1). The 11 selected m/z shown in Figure 1 have the highest loading in each group and are labeled in the loading plot as shown below. The number of found correspondences against k for different δ is also shown in the figure (figure 2).

Figure 1. m/z found in mode 2 of PC1.

Figure 2. Number of correspondences against k .

As shown in the histogram in figure 2, the trend of the number of correspondences against k does not change with δ up to 50 ppm (only the absolute values increase as the search interval increases), and this is considered to be a good test of the robustness of this approach (in the main text of the revised version of the paper we only show $\delta = 15$ ppm). As discussed above, this threshold guarantees certain attributions with the limited number of elements in the samples while allowing enough selected datapoints to build a statistical distribution.

	$C_{12}H_8^+$	$C_{13}H_7^+$	$C_{13}H_9^+$	$C_{14}H_8^+$	$C_{14}H_{10}^+$	$C_{15}H_9^+$	$C_{15}H_{11}^+$	$C_{16}H_{10}^+$	$C_{16}H_{12}^+$	$C_{17}H_{11}^+$	$C_{18}H_{12}^+$	$C_{20}H_{12}^+$
$C_{12}H_8^+$												
$C_{13}H_7^+$		0										
$C_{13}H_9^+$												
$C_{14}H_8^+$	2	0	2	2								
$C_{14}H_{10}^+$		2		4								
$C_{15}H_9^+$	2			2	4							
$C_{15}H_{11}^+$	4			4								
$C_{16}H_{10}^+$	4	2	4	2		4	6	4				
$C_{16}H_{12}^+$	6	4	6	4		6	8	6	8			
$C_{17}H_{11}^+$				4								
$C_{18}H_{12}^+$	4	4	6	4	6	4		6	8	6	6	
$C_{20}H_{12}^+$	4	2		4	4	6		6				

Table 1. Individual correspondences found between the m/z identified by PC1 positive. The color code represents the value of k .

The detailed list of the individual correspondences can be found in Table 1 above: similarly to Table 1 in the original paper, the cell located at the intersection of each row and column represents the potential correspondence between couples of ions in mode 1 (listed in the first row and in the first column). Empty cells represent no correspondences found. The numbers in the cells are the k values to match the closest ion in mode 2 found as explained above (15 ppm, only the main ion of each group of loadings). In this revised version of the table, the color code only represents the value of k and not the “quality” of the assignment as in the original table. *It is important to notice that this improved data processing protocol does not change the general conclusions of the paper.* However, it clearly shows that the original data processing protocol was insufficient for a complete overview of the space of the possible combination of ions in mode 1. Because of the improved data processing protocol, $[H]_{\text{TOF-SIMS}}$ of the selected m/z in mode 2 narrows down and slightly shifts to lower values (from 0.34-0.40 down to 0.33-0.37). This is due

to the large number of matches found for $k = 4, 6$. The main text in the revised version of the paper has been updated to account for the new data processing protocol, and the new section in the Methods *Analysis of the positive loadings of PC1* has been added to detail the loadings selection protocol. New trends become visible and are now discussed in the main text.

- $k = 0$, direct matches. We do not yet have a conclusive explanation for the detection of direct matches. Similarly to the case of covalent dimers, we estimate the formation of van der Waals dimers during the ToF-SIMS analysis unlikely: as discussed in the Methods section *Is the clustering of PAHs representative of flame processes?*, unrealistically high concentrations of monomers are required to generate clusters in the plume. On the other hand, morphological information on the core of soot particles show partially ordered structures that suggest stacking interactions between aromatic molecules, therefore van der Waals structures *might* turn into long-lived species (enough to be detected) when encased on NSPs. See for instance Kholghy et al., Carbon 100 (2016) 508-536.
- $k = 1, 3, 5, 7$, odd k values. In our opinion, one of the most interesting features of Figure 2 is that no correspondences between ions in mode 1 and mode 2 are found for odd k values. In the specific case of $k = 1$, as the Reviewer noted, the elimination of 1H might result in short-lived species. However, much depends on the structure of the molecular ion: in some cases, for instance if a methylene group in a 5-member ring is next to an aromatic system, post-ionization H-elimination can generate a resonantly-stabilized radical cation during the ionization step.
- $k = 2, 4, 6, 8$, even k values. These correspondences *can* be explained with the formation of covalent bonds from pre-existing molecular species. Arguments supporting this interpretation are: (1) as discussed in the introduction of the paper, including in theoretical models of soot formation the (reversible) dimerization of small PAHs followed by the stabilization of the dimer through the formation of C-C covalent bonds considerably improved calculations of the soot volume fraction. These covalent dimers would be particularly stable and thus "easier" to detect, see for instance but not only: Kholghy et al., Phys. Chem. Chem. Phys. 20 (2018) 10926–10938, Keller et al., Energ. Fuels 33 (2019) 10255–10266, Frenklach and Mebel, Phys. Chem. Chem. Phys. 22 (2020) 5314–5331, Martin et al., J. Phys. Chem. C 123 (2019) 26673-26682. (2) In this work, PCA finds a positive correlation between the signals of mode 1 and mode 2, then several correspondences are found between *couples* of m/z of mode 1 and *individual* m/z of mode 2. (3) The 2H-elimination is consistent with the hypothetical formation of one C-C covalent bond, thus $k = 2, 4, 6, 8$ would correspond to the formation of 1, 2, 3, 4 covalent bonds. (4) Finally, tests on pure PAHs and PAHs adsorbed on black carbon (please see the Methods section of this paper) show that clustering in the plume requires unrealistically high concentrations of monomers and thus exclude analytical artifacts during the analysis.

According to the above discussion, please note that the main text has been revised, and Table 1 has been replaced with a merged figure to provide a global view of the whole investigated range of k values rather than the three specific cases discussed in the original paper.

Comment 5. There are no matches found for "correspondences -1H ", could it be because such structures are very short lived? Because to form -2H matches from directs match there is large energy barrier for simultaneous loss (abstraction) of two hydrogen atoms.

Answer. The answer to this comment is strictly related to the answers to Reviewer1, comments (4) and (6) and Reviewer3, comment (1), and therefore one comprehensive answer is given. Please see answer to Reviewer1, comment (4).

Comment 6. In table 1, for "direct match" mostly ions with up to 16 carbon atoms are found, however, the map for -2H matches is wider and even ions with 20 carbon atoms are found. Why is

that? Could it be because direct matches for larger ions have lower concentration but covalently bounded matches are more stable and more likely to be found? Is it possible to comment on the relative concentration of direct matches and those with -2H matches?

Answer. The answer to this comment is strictly related to the answers to Reviewer1, comment (4) and (5) and Reviewer3, comment (1), and therefore one comprehensive answer is given. Please see answer to Reviewer1, comment (4).

Comment 7. "355 peaks (59%) are identified with high confidence (< 5 ppm) ", is 5 ppm the uncertainty in the concentrations?

Answer. The mass relative uncertainty given here in ppm is an estimation of the confidence given to the attributions of molecular formulae to the experimentally measured m/z as $\delta = \text{abs}(m/z - M_r)/m/z$ and it is not related to the concentrations (also usually expressed in ppm). Please see answer to Reviewer1, comment (4) for more details.

Comment 8. I do not understand the logic behind Figure 3. Also, I have difficulty understanding how estimating "La" helps us reduce possible contributors to soot as all combinations are below the 1 nm threshold.

Answer. The idea behind Figure 3 (Figure 4 in the revised paper) is to show that several important variables like [H], [C] and [O] can be calculated from ToF-SIMS data by using the same approach that Casiraghi et al. and Buijsters et al. used for hydrogenated carbon films. The analysis of the data in Figure 3 reveals that the linearity of $[\text{H}]_{\text{ToF-SIMS}}$ against $\log m/I(\text{G})$ is preserved for soot samples. For a detailed discussion on the limits of this approach please see the answer to Reviewer2, comment (8).

The idea of using the size of the aromatic island L_a to narrow down the possible structural formulae of PAHs was initially thought for the full investigated m/z range that includes high m/z for which structures above 1 nm are possible. In any case, L_a still provides useful information as it confirms that aromatic structures having size around 1 nm exist in the analyzed samples. This point has been corrected in the revised version of the paper.

Reviewer #2

The paper addresses an area of significant importance with both scientific and practical implications and as such is likely to be of interest to readers. In the main the paper is reasonably well written with some interesting findings, e.g. the confirmation of the absence of soot particles in the condensable "halo" material and the approximate bimodality of the in m/z prior to soot formation. The discussion regarding Fig. 2 is also excellent, clear and so on. However, some revision is required prior to publication with main issues below.

Comment 1. The discussion on thermodynamically-controlled soot formation makes two omissions that must be corrected. Firstly, the work by the Stanford group of Wang (e.g. Wang, Formation of nascent soot and other condensed-phase materials in flames, Proc. Combust. Inst. 33(1) (2011) 41-67) suggests entropy as a driving force through dehydrogenation reactions. Secondly, there are significant uncertainties in the determination of thermodynamic data for large PAHs as discussed by the Imperial College group (e.g. Lindstedt and Waldheim, Modeling of soot particle size distributions in premixed stagnation flow flames, Proc. Combust. Inst. 34(1) (2013) 1861-1868) that needs to be recognised in the context of reversibility. The latter work also shows that soot PSDs can be computed accurately starting with dimerization, while also recognising the need for work of the current kind due to the necessary transition from PAH dimerization to the van der Waals enhanced collision regime.

Answer. We thank the Reviewer for providing some literature that we did not cite in the context of thermodynamically-controlled soot formation. The paper Proc. Combust. Inst. 33 (2011) 41-67 has been re-cited in a new dedicated sentence, and the paper Lindstedt and Waldheim, Proc. Combust. Inst. 34 (2013) 1861-1868 has been added to the discussion on the modeling of NSPs size distribution, both in the introduction. The uncertainty on the determination of PAHs thermodynamic data is also briefly mentioned in the introduction.

Comment 2. The role of radical reactions is also highlighted in the work mentioned under point 1 and the importance of C5 ring structures in PAHs is relevant in this context as, for example, some resonantly stabilised radicals (e.g. indenyl) become prevalent in flames. (e.g. Lindstedt, Chemical Complexities of Flames, Proc. Combust. Inst. 27 (1998) 269-285). Also see the related discussion on p. 11-12 in the current paper. As stated, there has been recent further corroboration of such observations. However, the current paper should be amended in this regard to recognise that the issue was established quite some time ago.

Answer. We thank the Reviewer for suggesting these interesting papers. We would like to point out, however, that the paper Lindstedt, Proc. Combust. Inst. 27 (1998) 269-285 focuses on the role of 5-member rings on the formation and chemical growth of PAHs before soot formation, not during the nucleation process as discussed in our paper.

Many loadings in both modes are compatible with resonantly-stabilized radicals (in practice, many of the molecular formulae having odd number of C atoms). Resonantly-stabilized radicals are important species in the chemistry of flames, and their role in soot formation is a currently much discussed topic in the literature, see for instance the recent works Johansson et al., Science 361 (2018) 997-1000 and Frenklach and Mebel, Phys. Chem. Chem. Phys. 22 (2020) 5314-5331. As some experimental evidence suggests, some radicals are long-lived and can persist in soot samples long enough to be detected by ex-situ techniques like electron paramagnetic resonance, see for instance Vitiello et al., Combust. Flame 205 (2019) 286-294.

Comment 3. The use of abbreviations is very prevalent. This is fine. However, PC2 and PC1 are introduced in a haphazard way on p. 4 with essential information relegated to Fig. 1 (e.g. what PC1 >

0 stands for). The authors should correct such matters to make the paper more readable. The data presented is, however, interesting.

Answer. The beginning of the Results section has been extended and partially re-organized to be easier to read and understand without reading the Methods section. A brief description of the PCA is also integrated in the text together with a description of the data interpretation process, and in the Methods section ToF-SIMS and principal component analysis.

Comment 4. There appears to be an error in Fig. 1(d) where both the Impaction ROI and Halo ROI are defined by $PC2 < 0$. Certainly, the text suggests a different interpretation or perhaps the writing is in need of attention. The 19% PC2 result is poor and needs to be commented on further. By contrast, the PC1 value of 70% is quite impressive.

Answer. The impaction ROI is supposed to correspond to $PC2 > 0$. This error in Figure 1d has been corrected. The discussed values represent the percentage of the variance explained by each principal component. In our case, a 2D representation of the ensemble of the mass spectra that uses only PC1 and PC2 explains the $70+19=89\%$ of the total variance. In other words, in our discussion we trade a complex representation in 601 dimensions (the number of m/z used as variables) that contains 100% of the information on the variance for a much simpler representation in 2 dimensions (PC1 and PC2, obtained as linear combinations of the 601 m/z) that contains already 89% of the information on the variance. Since PC1 could be associated to the HAB and PC2 to the ROI, we are sure in our claim that the variability of the mass spectra is mostly explained by the change of HAB (70%) with a relatively minor contribution of the ROI (19%). 11% of the variance remains unexplained in this interpretation.

Comment 5. In terms of the Halo and Impaction ROIs, the paper fails to make reference to the earlier work of Abid et al. (e.g. Abid, Heinz, Tolmachoff, Phares, Campbell, Wang, *Combust. Flame* 154 (2008) 775–788 and Abid, Tolmachoff, Phares, Wang, Liu, Laskin, *Proc. Combust. Inst.* 32 (2009) 681–688). This should also be corrected.

Answer. The paper Abid et al., *Combust. Flame* 154 (2008) 775-788 is an important reference for the development of the sampling system used in this work, and as such is cited in our previous work Irimiea C. et al., *Carbon* 144 (2019) 815-830. However, the discussion in Abid et al., *Combust. Flame* 154 (2008) 775-788 is not considered pertinent to this work as it focuses on the description of a single particle impaction (the halo observed by the Authors in AFM analyses is attributed to the splashing of a single liquid-like soot particle). Conversely, in this work a large number of particles are impacted and the halo is generated by the diffusion of condensable gas far from the impaction region.

Comment 6. The top of p. 6 reads "...clusters. 45, 50 and 55 mm HAB.." This is not an acceptable way of writing. Please correct. The valuable technical conclusion that there is a change in the chemical composition downstream in the flame is credible, but also obscured by the writing.

Answer. This section has been heavily updated to take into account this comment and the similar remark from Reviewer1, comment (2).

Comment 7. On p. 8 it is argued that artifacts due to sampling and analysis are insignificant. This is a key point in the paper and the reference should be made correctly to the methods section.

Answer. Corrected.

Comment 8. The global fraction of [H] content is very interesting and well-performed. While the R2 norm is excellent, there are large uncertainties and it is not stated why (the very credible observation) that -O containing species could be a cause and, if so, to what an extent. It is accepted that this is speculation, but it should be more firmly grounded.

Answer. The uncertainties shown in Figure 3 (Figure 4 in the revised paper) represent the random error and are calculated from the standard deviation of three independent measurements then propagated using an approximation to a first-order Taylor series expansion for $[H]_{\text{Raman}}$ and $[H]_{\text{ToF-SIMS}}$. Conversely, the systematic error introduced by the existence in the peak list of unknown species (peaks used for the PCA but not identified) is tricky to estimate since, if a molecular formula is not known, it cannot be included in the calculation of $[H]_{\text{ToF-SIMS}}$. The ion count on the selected peak list of unknown species varies from a minimum of 19.5% at 45 mm HAB up to a maximum of 27.4% at 60 and 65 mm HAB (the average on all mass spectra at all HABs and ROIs is close to 22%). Although it might be tempting to use these values as a rough estimation of the confidence given to $[H]_{\text{ToF-SIMS}}$, if some speculation is accepted a better estimation can be based on the mass defect analysis as it follows. In the mass defect plot shown in Supplementary Information 2, it can be seen that the unknown species are mostly located in region (4) below the main sequence of PAHs in region (3). Their mass defect Δ is very low, slightly positive but close to zero even at high m/z . Assuming H, C and O to be the main contributors to the chemical composition of these species, we can imagine two scenarios consistent with the experimentally observed $\Delta \approx 0$:

- A. H-poor carbon clusters: $\Delta(\text{C})=0$ so any number of C in a cluster results in $\Delta=0$, and $\Delta(\text{H})>0$;
- B. H-rich, O-rich species: $\Delta(\text{H})>0$ compensate for $\Delta(\text{O})<0$. Additionally, since $\Delta(\text{H})=+0.0078$ and $\Delta(\text{O})=-0.0051$, to keep $\Delta=0$ the ratio between H and O should be approximately constant at 2H:3O.

In order to obtain a rough estimation of the systematic error, we tentatively assigned molecular formulae to the remaining unknown peaks in the frame of scenarios A and B then recalculated $[H]_{\text{ToF-SIMS}}$ in these two extreme cases. However, it is important to notice that some of these assigned molecular formulae are quite unrealistic: the peaks showing a large mismatch are most likely the result of the convolution of more than one contribution that unfortunately cannot be resolved with the currently available mass spectrometer. Keeping in mind the equation for calculating $[H]_{\text{ToF-SIMS}}$:

$$[H]_{\text{ToF-SIMS}} = \frac{N_{\text{H}}}{N_{\text{H}} + N_{\text{C}} + N_{\text{O}}} \quad \text{with} \quad N_{\text{X}} = \sum_i N_{\text{X},i} w_i \quad \text{and} \quad \sum_i w_i = 1$$

it can be seen that, since the overall low value of N_{O} in the collected soot samples imposes a similarly low value of N_{H} in both scenarios, while not affecting much N_{C} ($N_{\text{O}} \ll N_{\text{C}}$), in practice both scenarios result in close $[H]_{\text{ToF-SIMS}}$ around 10% lower than the values shown in Figure 3 (Figure 4 in the revised paper).

In conclusion, to obtain a more accurate estimation of the real $[H]_{\text{ToF-SIMS}}$, as many as possible of the unknown species have to be identified, however this requires a higher resolving power instrument that unfortunately is not available at this time. $[H]_{\text{ToF-SIMS}}$ currently calculated in the paper is affected by a systematic error due to neglecting the contributions of unknown species, and overestimates the real $[H]_{\text{ToF-SIMS}}$ by 10% at most. This estimation is based on the mass defect analysis, but requires attributing tentative molecular formulae to a minor fraction ($\frac{1}{4}$ at most) of the total ion count, and therefore it should be used with prudence and restricted to an extreme limit inferior. A dedicated discussion has been added to the Methods section *Eq. 2 sensitivity analysis*.

Comment 9. The discussion on p. 12 concerning vdW forces should usefully also refer to point 1 above.

Answer. Please see the answer to Reviewer1, comment (4).

The subsequent discussion is fine and the methods discussion shows the commendable care taken by the authors. The expectation is that the above revisions can readily implemented and that this would result in a paper suitable for publication.

Reviewer #3

The paper studies the dimerization of PAH which is thought to be the source of soot nucleation but much is unknown. The possibility of dimers of low mass PAH resulting in soot nucleation has been much debated but to date there has not been experimental confirmation. The paper provides a novel and convincing argument supported by detailed experiments that dimers of low-mass PAH occur just before soot nucleation and supports the reactive dimerization model. The paper provides a detailed discussion of the chance that the dimers have been formed in the probe, on the wafer or on soot particles rather than in the gas phase. This paper marks significant progress in our understanding of soot nucleation and will be highly influential to the field.

Detailed Comments:

Comment 1. On pages 7 and 18, you say that stable covalent bonds cannot be formed during sampling; but could dimers formed by Vander Waals forces be possible in the sampling system?

Answer. The answer to this comment is strictly related to the answers to Reviewer1, comments (4), (5) and (6). Please see answer to Reviewer1, comment (4). Briefly, as discussed in the Methods section *Is the clustering of PAHs representative of flame processes?*, clustering in the sampling line or in the ion source of the mass spectrometer requires unrealistically high concentrations of monomers, therefore we do not believe it to be the origin of the observed matches.

Comment 2. Do all data points in figure 1c correspond to $PC_2 > 0$? If so, you should clarify. Why do both the halo and impaction zone of PC_2 (figure 1d) correspond to $PC_2 < 0$? This seems to contradict the statement at the bottom of page 4.

Answer. The impaction ROI is supposed to correspond to $PC_2 > 0$. This error in Figure 1d has been corrected.

REVIEWERS' COMMENTS:

Reviewer #1 (Remarks to the Author):

The authors have thoroughly answered my questions and implemented many of the suggestions. Thus, I recommend the paper for publication.

Reviewer #2 (Remarks to the Author):

I am satisfied that the revised version meets the suggested revisions. I would have preferred that the authors cite the work by Abid et al. to provide a better balance of references. However, this is not a requirement. The current paper is enjoyable to read and suitably novel.

Reviewer #3 (Remarks to the Author):

All my comments have been addressed. I supported accepting the paper.